# LM-Infinite: Simple On-the-Fly Length Generalization for Large Language Models

## Abstract

In recent years, there have been remarkable advancements in the performance of Transformer-based Large Language Models (LLMs) across various domains. As these LLMs are deployed for increasingly complex domains, they often face the need to follow longer user prompts or generate longer texts. In these situations, the *length generalization failure* of LLMs on long sequences becomes more prominent. Most pre-training schemes truncate training sequences to a fixed length. LLMs often struggle to generate fluent and coherent texts after longer contexts, even with relative positional encoding specifically designed to cope with this problem. Common solutions such as finetuning on longer corpora often involve daunting hardware and time costs and require careful training process design. To more efficiently extrapolate existing LLMs' generation quality to longer texts, we theoretically and empirically investigate the main out-of-distribution (OOD) factors contributing to this problem. Inspired by this diagnosis, we propose a simple yet effective solution for on-the-fly length generalization, LM-Infinite. It involves only a $\Lambda$-shaped attention mask (to avoid excessive attended tokens) and a distance limit (to avoid unseen distances) while requiring no parameter updates or learning. We find it applicable to a variety of LLMs using relative-position encoding methods. LM-Infinite is computationally efficient with $O(n)$ time and space, and demonstrates consistent text generation fluency and quality to as long as 128k tokens on ArXiv and OpenWebText2 datasets, with 2.72x decoding speedup. We will make the codes publicly available following publication.

## 1 Introduction

The evolution of Natural Language Generation (NLG) in recent years has been significantly driven by the progress of Large Language Models (LLMs) (Wei et al., 2022a; Kojima et al.; Wei et al., 2022b; Brown et al., 2020; Li et al., 2023b). LLMs have been successfully applied to a wide variety of tasks, demonstrating an impressive ability to understand and generate natural language across different contexts, such as Document Understanding, Information Extraction, and Cross-document Question Answering, Bai et al. (2023)

However, as LLMs are increasingly used in complete scenarios such as long document generation and long dialogue systems, LLMs face the challenge of *length generalization failures* on long text sequences. Despite extensive explorations in smaller-scale models (Press et al., 2021; Sun et al., 2022; Chi et al., 2023), current state-of-the-art (SoTA) LLMs still struggle to directly generalize to sequences of unseen lengths. When forced to generate after too long contexts, they usually compromise the generation fluency. In most pre-training schemes, to control the exploding time and economic costs of long text lengths, practitioners have to bound training lengths, such as 2048 tokens for LLaMA (Touvron et al., 2023a). When there is a gap between training and inference lengths, LLMs fail to recognize the input and start to generate gibberish, despite the use of advanced techniques such as relative position encoding which were proposed to deal with this problem. Numerous relative position encoding techniques such as RoPE(Su et al., 2021) and Alibi(Press et al., 2021) have been widely adopted by state-of-the-art LLMs. The main idea behind relative position encoding is that, instead of using absolute position information of tokens, the attention weight between two tokens relies on their distance in sequence. These designs are theoretically capable of running on unseen lengths, but on LLMs, we still observe generalization failures or Not-a-Number (NaN) values on inputs longer than training time (Kaiokendev, 2023) (see also Sections 3 and

5). Besides, the $O(n^2)$ computational complexity of the prevailing Transformer-based LLMs also means overwhelming hardware and financial demand.

Inspired by the length generalization mystery, we undertake an empirical investigation of the main factors contributing to this generalization failure problem. In Section 3 through theoretical and empirical analyses, we identify three out-of-distribution (OOD) factors: *unseen distances*, *unseen number of tokens under attention*, and *implicitly encoded positional information*. Building upon these findings, we propose LM-Infinite, a surprisingly simple yet efficient solution compatible with various LLMs that use relative position encodings. LM-Infinite introduces two innovative elements: a $\Lambda$-shaped attention mask and a distance bound during attention. As important advantages, it does not require any parameter updates for pre-trained LLMs and only involves $O(n)$ computational complexity. LM-Infinite also provides a 3.16x speedup on encoding and 2.72x speedup on decoding.

Empirically, LM-Infinite demonstrates generalizability to sequences of much longer lengths, capable of maintaining consistent fluency and generation quality on documents with as many as 128k tokens in ArXiv (academic preprints) and OpenWebText2 (Reddit submissions) for a wide range of SoTA LLMs: LLaMA (Touvron et al., 2023a), Llama-2 (Touvron et al., 2023b), MPT-7B (Team, 2023) and GPT-J (Wang & Komatsuzaki, 2021). It achieves performance superior or comparable to LLMs explicitly fine-tuned on long sequences, despite requiring no extra learning or parameter updates. In summary, our contributions in this work include:

- We analyze a behavioral model of LLMs regarding sequences longer than training time through theoretical and empirical diagnoses and explain multiple factors that contribute to LLMs' generalization failures.
- We propose a simple on-the-fly decoding method, LM-Infinite, which brings computational efficiency as well as generalizability to unseen lengths. This saves researchers from the cost of fine-tuning or even training from scratch.
- We conduct experimental evaluations of LM-Infinite. LLMs' fluency and generation quality are consistently maintained over 32k-length sequences on the ArXiv dataset, much longer than training time.

## 2 RELATED WORK

### 2.1 POSITIONAL ENCODINGS IN TRANSFORMERS

Since the advent of the Transformer (Vaswani et al., 2017), along with its variants (generally named Transformers), has become the most widely used architecture of modern LLMs, thanks to its performance and ability for parallel training. As the attention mechanism (the core component in Transformers) operates on a bag of token features regardless of their positions, Transformers usually rely on explicit designs to incorporate position information. These designs are called *positional encodings*, and can generally be categorized into two classes. The **absolute positional encodings** are those providing the absolute positions, usually with the help of a sequence of vectors called *position embeddings*. Examples of such include sinusoidal position embeddings added to the input token embeddings (Vaswani et al., 2017), or learned position embeddings in BERT (Kenton & Toutanova, 2019), or adding the dot product between two tokens' position embeddings on the attention logit (Ke et al., 2020). Recently, to overcome the drawback that Transformers become unfamiliar with unseen positions, **relative positional encodings** are proposed to use distance information between tokens instead. Such information is usually incorporated into attention layers. Examples include a learnable attention logit bias in T5 (Raffel et al., 2020), Transformer-XL Dai et al. (2019) and Sandwich (Chi et al., 2023), a fixed linear attention decay called Alibi Press et al. (2021), and rotating query and key sequences based on distances such as RoPE (Su et al., 2021; Li et al., 2023a), CAPE Likhomanenko et al. (2021) and XPos (Sun et al., 2022; Ding et al., 2023). XPos and Longformer (Beltagy et al., 2020) also propose a block-diagonal attention mask, which however relies on explicit training to familiarize LLMs with it, and is not a plug-and-play tool like ours. As an extreme example, NoPE (Kazemnejad et al., 2023) claims that the Transformer can implicitly encode positional information, so no positional encoding is needed. Despite theoretical promises and experimental verification on smaller scale experiments in these papers, length generalization failures are still widely observed when directly applied to large language models (Kaiokendev, 2023). This gap motivates us to hypothesize that there still exist OOD factors in relative positional encoding. In Section 3

we identify such factors and demonstrate that removing them allows relative positional encoding to have perfect length generalization on extremely long sequences.

## 2.2 FINE-TUNING ON LONGER TEXTS

In light of generalization failures observed in LLMs, one straightforward solution is to finetune LLMs on longer text sequences, so that unseen positions can be exposed to LLMs for familiarity. Chen et al. (2023) interpolate positional encoding on longer sequences for finetuning. Tworkowski et al. (2023) adopt contrastive learning while finetuning on longer texts. Tao et al. (2023) and Kiyono et al. (2021) use padding and shifting for synthesizing long texts, respectively. These temporary remedies push the context length limit further but do not address the root causes of length generalization failures. They also require massive training resources due to the large sizes of LLMs. In contrast, our work aims at an on-the-fly solution by diagnosing the OOD factors preventing length generalization, and greatly saving resource costs.

## 2.3 OTHER EFFORTS TOWARDS LONG-CONTEXT LLMS

Besides directly addressing the length generalization problem, other solutions are proposed to grant LLMs access to longer contexts without really reading them in full. For example, Recurrent-GPT (Zhou et al., 2023) prompts an LLM to recurrently generate texts, while at each iteration only reading the most recent context and a summary of longer histories. Some other work introduces special mark-up tokens (Bueno et al., 2022) or landmark tokens (Mohtashami & Jaggi, 2023) that allow LLMs to access a subset of most informative tokens. Anil et al. (2022) propose a prompting strategy that, when combined with pre-training and fine-tuning, is able to generalize to unseen lengths. Besides, Yang et al. (2023) use an outliner and a controller for two-staged long story generation. Finally, augmenting LLMs with retrieval-based memories Wu et al. (2021); Guu et al. (2020); Borgeaud et al. (2022); Khandelwal et al. (2019); Kaiser et al. (2016); Yogatama et al. (2021) lets LLMs only read retrieved information from a large database. These designs, however, usually need explicit finetuning and are not directly compatible with the existing state-of-the-art LLMs. Our work, in contrast, aims at extending *existing* LLMs to longer texts on the fly, which better leverages their impressive generalization power.

## 3 DIAGNOSING OOD FACTORS IN LLMS

In this section, we diagnose the out-of-distribution (OOD) factors contributing to the length generalization failure. We analyze with both theoretical analysis and experimental verification.

We are mainly inspired by the hypothesis that relative positional encodings in pre-trained LLMs already capture the ability to deal with relative positions. However, when applied to longer sequences, the internal features (such as attention weights and hidden states) become "unfamiliar" to LLMs, i.e., out of the training distribution. Upon removal of these factors, we might shift internal features back to the training distribution, which are "comfort zones" to LLMs. Therefore LLMs will be able to generate with their original quality. In this section, we search for such factors. The intuition is to look for internal features that might be OOD and verify their existence.

### 3.1 OOD FACTOR 1: UNSEEN DISTANCES

Recall that, in relative positional encoding, the attention weight between two tokens depends on their distance. It is intuitive to realize that, if texts become too long, some distance values will increase to an unseen large number, eventually exceeding those seen in pre-training. In the following, we will demonstrate formally and empirically that, as length increases, the attention logits will have to explode to infinity for the attention functions to distinguish new distance values.

Let us denote the attention function in a relative position encoding as $w(\mathbf{q}, \mathbf{k}, d) \in \mathbb{R}$. Here $w(\cdot, \cdot, \cdot)$ takes the query vector $\mathbf{q}$, the key vector $\mathbf{k}$ and their distance $d$, and returns a scalar as attention logit. The final attention weights are usually calculated by a softmax operation. Specifically, if there are $n$ tokens with indices $(1, \cdots, n)$, the attention paid by the last token on a preceding token $i$ is

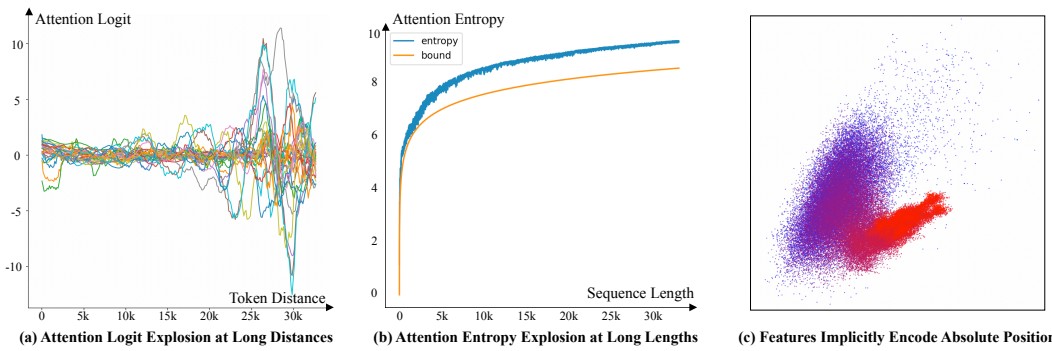

Figure 1: Diagnosis of three OOD factors in LLMs.

calculated as

$$\text{Attn}(\text{token}_n, \text{token}_i) = \frac{e^{w(\mathbf{q}_n, \mathbf{k}_i, n-i)}}{\sum_{j=1}^{n} e^{w(\mathbf{q}_n, \mathbf{k}_j, n-j)}} \quad (1)$$

**Theorem 1.** *(Long-Distance Attention Logit Explosion) Let* $\mathbf{q}$ *and* $\mathbf{k}$ *be random vectors from distributions* $\mathcal{D}_\mathbf{q}$ *and* $\mathcal{D}_\mathbf{k}$*, respectively. We use the pseudo-dimension* $\dim_P(\cdot)$ *defined in Pollard (1990), which measures the representation capacity of a function family. Assume that the set of distance-based logit functions* $\mathcal{H} = \{w(\cdot, \cdot, d) | d \in \mathbb{N}\}$ *has bounded pseudo-dimension* $\dim_P(\mathcal{H}) = r$[1]. *Let us also define the distinguish-ability of two distances* $d$ *and* $d'$ *under* $w$ *as their expected squared difference:* $\mu_w(d, d') = \mathbb{E}_{\mathbf{q} \sim \mathcal{D}_\mathbf{q}, \mathbf{k} \sim \mathcal{D}_\mathbf{k}} (w(\mathbf{q}, \mathbf{k}, d) - w(\mathbf{q}, \mathbf{k}, d'))^2$. *We assume that* $w$ *will not recognize only a finite group of distances, otherwise, all distances longer than a threshold will become almost the same as shorter distances. Formally, for any* $n$*, there is a partition of* $[0..n]$ *into* $\alpha(n)$ *groups so that,* $\mu_w(d, d') \leq \epsilon$ *for any* $d, d'$ *from the same group.* $\alpha(n) \in \mathbb{N}$ *is non-decreasing and unbounded function. Then we have:*

$$\sup_{\mathbf{q}, \mathbf{k}, d \leq n} |w(\mathbf{q}, \mathbf{k}, d)| \geq \left(\frac{\alpha(n)}{2}\right)^{\frac{1}{2r}} \frac{\epsilon}{4e}.$$

The proof can be found in Appendix A. We also empirically verify this on LLaMA on 32 sequences in the ArXiv dataset, truncated down to 32k tokens. We select the 0-th attention head in each Transformer layer for clarity of visualization, and plot the attention weights paid by the last token to all preceding tokens in Figure 1(a). We can see that at long distances, the attention logits oscillate to significantly larger absolute values than those within the training length of 4k.

The takeaway message is that either the attention logit functions $w(\cdot, \cdot, d)$ fail to recognize the unseen distances, or their values will increase to infinity. The latter case will lead to OOD logits, which are "unfamiliar" to LLMs, and potentially result in irregular results. Even if the former case is true, in the next Section we show that it will cause another type of OOD factor. To alleviate the current factor, we conjecture that *one needs to limit the distance values during attention computation.*

### 3.2 OOD FACTOR 2: UNSEEN NUMBER OF TOKENS

Another factor that potentially causes out-of-distribution is the number of tokens to attend to. When texts become longer, later tokens will need to attend to more tokens. This might dilute the attention weights and make the attention distribution more flattened, causing a loss of information in the attention. Here we study the entropy values, which is a theoretical metric for measuring the informativeness of a distribution. In the next proposition we formally demonstrate that, unless the logits explode, the entropy of attention weights will increase to infinity. In other words, there is a dilemma between the OOD factors 1 and 2.

---

[1]This is true for most current techniques. See discussions in Appendix C

**Proposition 1.** *(Attention Entropy Explosion) Let $w_1, w_2, \cdots, w_n \in [-B, B]$ be a sequence of attention logits. Then the entropy of the attention distribution will increase to infinity:*

$$Entropy\left(\left(\frac{e^{w_i}}{\sum_{j=1}^{n} e^{w_j}} \Big| 1 \leq i \leq n\right)\right) = \Omega(\ln n)$$

The proof is provided in Section B. We go on to empirically verify it in practice. We follow the setting in Section 3.1 and plot the attention entropy against context lengths in Figure 1(b). The curve indeed shows an ever-increasing attention entropy. We also contrast it with an estimated bound in Proposition 1.

This finding suggests a limit on the number of tokens to be attended to so that LLMs can operate on familiar attention distributions. After analyses of these two factors, one might be tempted to propose an easy solution: forcing each token only to attend to the nearest few tokens, ignoring all farther tokens during attention. This is similar to the block-diagonal attention mask used in XPos (Sun et al., 2022) and Longformer (Beltagy et al., 2020). However, we find that this does not work and LLMs' performance actually degrades on shorter texts. It means that XPos' extrapolation ability heavily relies on explicit training, and is not directly applicable to other LLMs. This phenomenon indicates the existence of another OOD factor, which we analyze in the following section.

### 3.3 OOD FACTOR 3: IMPLICITLY-ENCODED ABSOLUTE POSITION

In this section, we are going to demonstrate a counter-intuitive phenomenon. Even if absolute position information is not explicitly encoded in the computation graph, the attention mechanism is still able to implicitly encode it. We conjecture that this happens in Transformers with relative positional encodings. The following theorem from Kazemnejad et al. (2023) proves this fact:

**Theorem 2.** *(Implicitly Encoded Position) Let $x$ be an input sequence of length $T + 1$ without positional encoding. Then there exists a parameterization for a vanilla self-attention layer such that its output features are able to recover absolute positions $[1, ..., T + 1]$.*

In the construction provided in Kazemnejad et al. (2023), the starting tokens' signals are stronger and easier to distinguish than tailing tokens. If this is true, then it suggests another potential OOD factor. When the length is short, the LLM can implicitly encode positional information of initial tokens. However when the length exceeds those seen in the training corpus, initial tokens are mishandled due to OOD factors 1 and 2, and their absolute position information might become distorted or missing. However, the theorem is existential: it only proves that implicitly encoding absolute positions is *possible*, but does not guarantee that this is actually *happening* in real LLMs. As an empirical verification, we take the hidden states output by the first layer of LLaMA and plot a Principal Component Analysis (PCA) projection into a 2-d plane in Figure 1(c). The dots correspond to the first 4096 tokens in 32 sequences, with blue ones corresponding to the initial tokens and red tokens being the tail ones. In the plot, we see that tokens at different positions do occupy distinct sub-spaces in the features space, even without explicit implementation to encode absolute position information. This provides an explanation of why the simple solution mentioned at the end of Section 3.2 fails: when the sequence becomes long, directly limiting the attention window will eliminate initial tokens so that the feature sub-space they occupy will become invisible for attention. We conjecture that keeping these starting few tokens is important for LLMs to normally function.

After identifying these OOD factors, we claim that we have found missing pieces behind the length generalization problem. In the following, we propose our solution LM-Infinite in Section 4, and picture a conceptual model depicting how the relative position encoding works.

## 4 LM-INFINITE

### 4.1 GENERAL PRINCIPLES

Based on the analysis above, we propose our solution, LM-Infinite, which is a simple on-the-fly technique for length generalization on Transformer-based LLMs with relative positional encodings. LM-Infinite provides a set of high-level principles which is not limited to one single LLM.

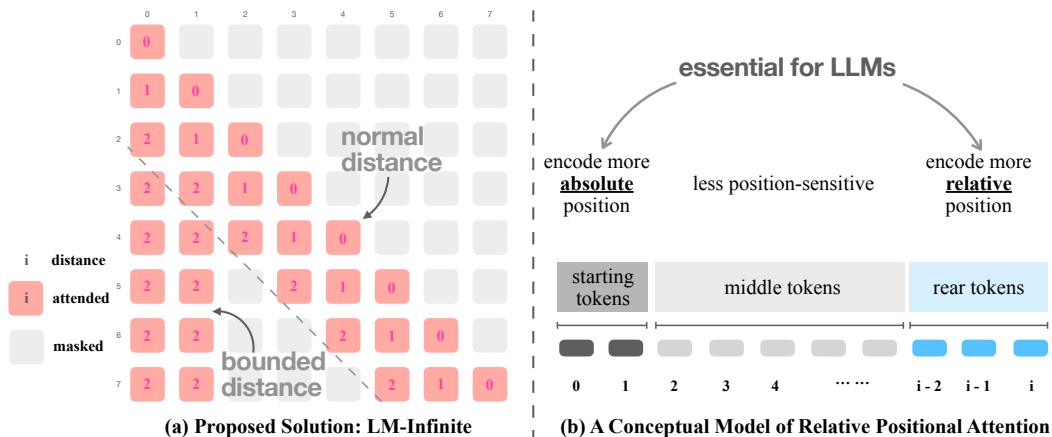

Figure 2: (a) LM-Infinite is a plug-and-play solution for various LLMs, consisting of a $\Lambda$-shaped mask and a distance constraint during attention. (b) We also provide a conceptual model for understanding how relative position encoding works.

An overview of LM-Infinite is illustrated in Figure2(a). This simple solution consists of two components: a $\Lambda$-shaped attention mask and a distance limit. As visualized in the figure, the **$\Lambda$-shaped attention mask** has two branches: a *global* branch on the left and a *local* branch on the right. The global branch allows each token to attend to the starting $n_{\text{global}}$ tokens if they appear before the current token. The local branch allows each token to attend to preceding tokens within $n_{\text{local}}$ distance. Any other tokens outside these two branches are ignored during attention. Here we heuristically set $n_{\text{local}} = L_{\text{pretrain}}$ as equal to the training length limit. This choice includes the "comfort zone" of LLMs in attention. The selection of $n_{\text{global}}$ has less effect on model performance, and we find that the range $[10, 100]$ is generally okay. Note that $n_{\text{global}} = 0$ will lead to immediate quality degradation. This design is based on the OOD factors 2 and 3 above, where we aim to control the number of tokens to be attended to, while also ensuring the inclusion of starting tokens. Theoretically, LM-Infinite can access information from a context as long as $n_{\text{layer}} L_{\text{pretrain}}$, because each deeper layer allows the attention to span $L_{\text{pretrain}}$ farther than the layer above.

The **distance limit** involves bounding the "effective distance" $d$ within $L_{\text{pretrain}}$. This only affects tokens that are in the global branch. In specific, recall that in relative positional encoding, the attention logit is originally calculated as $w(\mathbf{q}, \mathbf{k}, d)$, where $d$ is the distance between two tokens. Now we modify it as $w(\mathbf{q}, \mathbf{k}, \min(d, L_{\text{pretrain}}))$. This design is motivated by the OOD factor 1 and ensures that LLMs are not exposed to distances unseen during pre-training.

### 4.2 IMPLEMENTATION DETAILS

The principles in LM-Infinite are applicable to most relative positional encodings. As this work is focused on addressing the length generalization failure of LLMs, we will evaluate LM-Infinite on 3 families of SoTA open-sourced LLMs in Section 5: LLaMA series (LLaMA and Llama-2), MPT-7B series and GPT-J series. Both LLaMA and GPT-J use RoPE encoding, and MPT-7B uses Alibi encoding. The principles can be easily generalized to other relative positional encoding methods.

**RoPE** (Rotary Position Embedding) Su et al. (2021) proposes to rotate the key and query vectors based on positions before computing the inner product. Specifically, each vector $\mathbf{x}$ (either key or query) is split into pairs of elements $\{(x_0, x_1), (x_2, x_3), \cdots\}$, with each pair interpreted as a 2-d vector. RoPE then rotates the vector $(x_a, x_{a+1})$ of token $i$ with angle $\theta_{a,i} = i\omega_a$, where $\omega_a$ is the rotating speed associated with dimension pair $(a, a+1)$. After rotation, the 2-d vector becomes $\begin{pmatrix} \cos i\omega_a & -\sin i\omega_a \\ \sin i\omega_a & \cos i\omega_a \end{pmatrix} \begin{pmatrix} x_i \\ x_{i+1} \end{pmatrix}$. They show that the inner product between rotated $\mathbf{q}_i$ and rotated $\mathbf{k}_j$ is solely determined by values of $\mathbf{q}_i, \mathbf{k}_j$ and distance $|i-j|$. In LM-Infinite, the $\Lambda$-shaped mask is straightforward to implement on RoPE. For the limited distance principle, the local branch follows the original calculation. On the global branch (excluding the overlap with the local branch), we keep

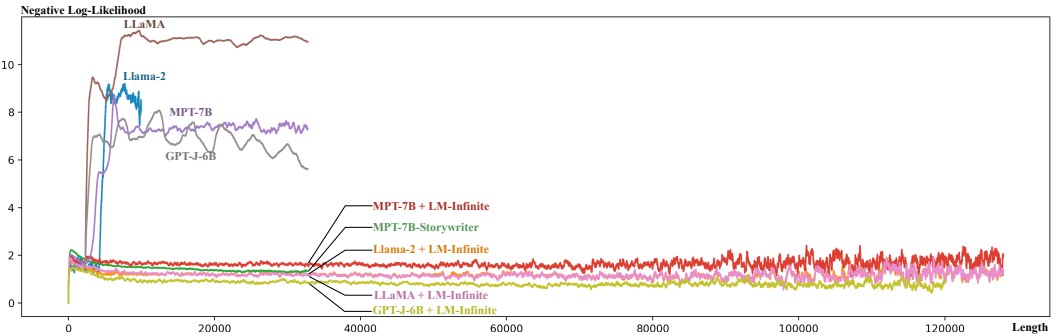

Figure 3: LM-Infinite flattens the NLL (negative log-likelihood, the logarithm of perplexity) curve of various LLMs on ArXiv dataset without any paramter updates. The trends are similar to MPT-7B-Storywriter, an explicitly fine-tuned LLM. Llama-2 outputs NaN values on long sequences so the curve is relatively shorter.

all $\mathbf{k}$ vectors unrotated and rotate all $\mathbf{q}$ vectors to a fixed distance $L_{\text{pretrain}}$. Then the two branches are composed together before attention masking.

**AliBi** Press et al. (2021) proposes to offset all attention logits between tokens $i, j$ by a linear term $-|m(i-j)|$ and become $\mathbf{q}_i^\top \mathbf{k}_j - |m(i-j)|$. To this end, the MPT-7B codes implement an offset matrix as an additive term in attention logits. To augment with LM-Infinite, we simply clip the offset matrix with a minimum value of $-|mL_{\text{pretrain}}|$.

### 4.3 A CONCEPTUAL MODEL FOR RELATIVE POSITION ATTENTION

In this section, we describe a conceptual model of how relative positional encoding functions in Figure 2(b), based on the OOD factor diagnoses and LM-Infinite designing principles. The figure illustrates the view when *generating one next token*, that is, the last token paying attention to all preceding tokens. In our conceptual model, a long context can be roughly partitioned into 3 parts:

1. The **starting tokens** encode predominantly their absolute position information as explained in Section 3.3. They are essential components for attention layers because their features occupy a specific region in the feature space (e.g., upper-right in Figure 1(c)). If this region is missing or attended to using an unseen large distance, this will create the OOD factor 3.

2. The **rear tokens** which are closest to the final token. Here the relative positions are more important. Rear tokens are essential for the attention layer to correctly function.

3. The **middle tokens** encode less position-sensitive information. As analyzed in Section 3.1 and 3.2, this region will either have exploding attention logits or too high attention entropy (OOD factor 1 and 2). Thus it does more harm than good for length generalization, so we remove them in LM-Infinite on sequences longer than training.

## 5 EVALUATION

In this section, we empirically evaluate LM-Infinite's performance. We select ArXiv and Open-WebText2 corpora from the Pile dataset (Gao et al., 2020), which consists of preprint papers from ArXiv and Reddit submissions, respectively. For LLMs to evaluate, we use LLaMA-7B, Llama-2-7b, MPT-7B, and GPT-J-6B. MPT-7B-Storywriter (fine-tuned on long sequences) is used as one of the baselines.

### 5.1 FLUENCY

We first evaluate the fluency of LM-Infinite using the widely adopted perplexity metric. Formally, when evaluating the quality of a probabilistic model $M$ on modeling a distribution $\mathcal{D}$, perplexity is defined as the exponentiation of average negative log-likelihood (NLL): $\text{PPL}(\mathcal{D}, M) =$

Table 1: Perplexity scores on ArXiv and OpenWebText2 dataset. LLMs with LM-Infinite achieve SoTA perplexity on 7 out of 9 columns while requiring no parameter updates.

| Model | Setting | ArXiv | | | | | OpenWebText2 | | | |
|---|---|---|---|---|---|---|---|---|---|---|
| | | 2k | 4k | 8k | 16k | 32k | 2k | 4k | 8k | 16k |
| Sandwich | Train | 5.02 | 5.15 | 5.28 | - | - | 23.3 | 23.8 | 24.7 | - |
| XPos | Train | 21.6 | 20.73 | - | - | - | - | - | - | - |
| LongLLaMA | Fine-tune | 8.17 | 7.44 | - | 6.94 | - | - | - | - | - |
| MPT-7B-SW | Fine-tune | 6.46 | 5.43 | 4.31 | 4.36 | 3.61 | 9.77 | 10.92 | 6.59 | **5.12** |
| MPT-7B | Vanilla | 5.49 | 247.6 | 1122 | 1672 | 1601 | 8.26 | 128.9 | 190.6 | 132.5 |
| LLaMA | Vanilla | 3.84 | 10k | 60k | 68k | 49k | 6.16 | 6636 | 456k | 44k |
| GPT-J-6B | Vanilla | 3.90 | 1285 | 1011 | 1617 | 278 | 8.83 | 746 | 1348 | 1803 |
| Llama-2 | Vanilla | **3.37** | 3.76 | 8461 | NaN | NaN | 6.18 | 5.76 | 6507 | NaN |
| MPT-7B | LM-Infinite | 5.69 | 6.76 | 5.79 | 5.98 | 4.60 | 8.46 | 12.25 | 8.54 | 8.93 |
| LLaMA | LM-Infinite | 4.38 | 4.54 | 3.68 | 4.20 | **1.02** | 6.33 | 6.08 | 9.53 | 7.03 |
| GPT-J-6B | LM-Infinite | 3.84 | **3.13** | **3.00** | **3.06** | 2.14 | 8.83 | 8.49 | **6.49** | 7.39 |
| Llama-2 | LM-Infinite | 4.33 | 3.63 | 3.33 | 4.18 | 6.49 | **6.13** | **5.32** | 8.28 | 8.15 |

$\exp(-\mathbb{E}_{x \in \mathcal{D}} \ln M(x))$. We plot the NLL curve in Figure 3 on the ArXiv dataset. Note that Llama-2 outputs NaN on too long sequences so the curve is relatively shorter. All vanilla models run out of memory at ∼32k lengths. We see that LM-Infinite successfully flattens the perplexity curve to lengths much longer than their training input lengths. This suggests a consistent and unharmed fluency in long sequences. The longer ends of curves have larger variances because of fewer documents with those lengths. Figure 5 provides an ablation study of why both components in LM-Infinite are essential for maintaining LLM functionality.

We also numerically log the perplexity scores at a few milestone lengths (2k, 4k, 8k, 16k, and 32k) on ArXiv and OpenWebText2 in Table 1, which shows a similar trend. Open-WebText2 has very few data over length 32k so we omit the column. Note that with the help of LM-Infinite, LLMs successfully accomplish length generalization and achieve SoTA perplexity scores in 7 out of 9 columns. This is an encouraging result considering that LM-Infinite does not require any parameter updates in contrast to numerous strong baselines. As a direct comparison, MPT-7B+LM-Infinite achieves only slightly inferior scores than its fine-tuned cousin, MPT-7B-Storywriter. This suggests that LM-Infinite is an efficient counterpart to resource-consuming fine-tuning.

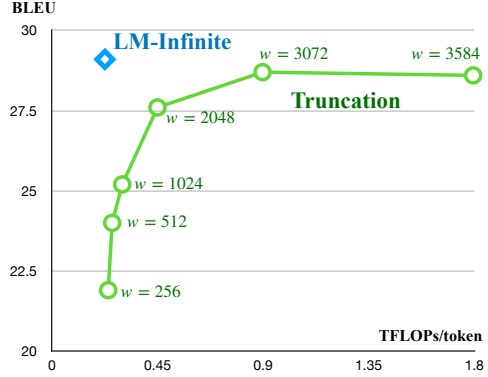

Figure 4: LM-Infinite achieves better balance between computation complexity with generation quality than simple tuncation.

## 5.2 GENERATION PERFORMANCE

As perplexity is an internal metric for LLMs, we evaluate LM-Infinite's generation quality on ArXiv and OpenWebText2 test sets, with BLEU (Papineni et al., 2002) and ROUGE (Lin, 2004) (ROUGE-LSum to be specific) as metrics. In simple words, both metrics evaluate the overlap on $n$-grams between the generated texts and the reference texts, while BLEU emphasizes on precision and ROUGE focuses on recall. We let the LLMs generate 100 tokens after each milestone length, and use the following 100 tokens in original texts as reference. As the generation takes a long time we sample 100 long sequences for evaluation in each dataset. The results are listed in Table 2. With a similar trend as the last section, LM-Infinite successfully allows LLMs to extend their generation quality to lengths longer than training, comparable to the effect of fine-tuning without parameter updates. Note that LM-Infinite has slightly different effects on different LLMs. On LLaMA and GPT-J-6B, the quality is better maintained at longer positions, while on Llama-2 the quality is better at nearer positions. We also evaluate the computation efficiency on a length of 32k in Appendix D, where LM-Infinite demonstrates a 3.16x speedup on encoding and a 2.72x speedup on decoding.

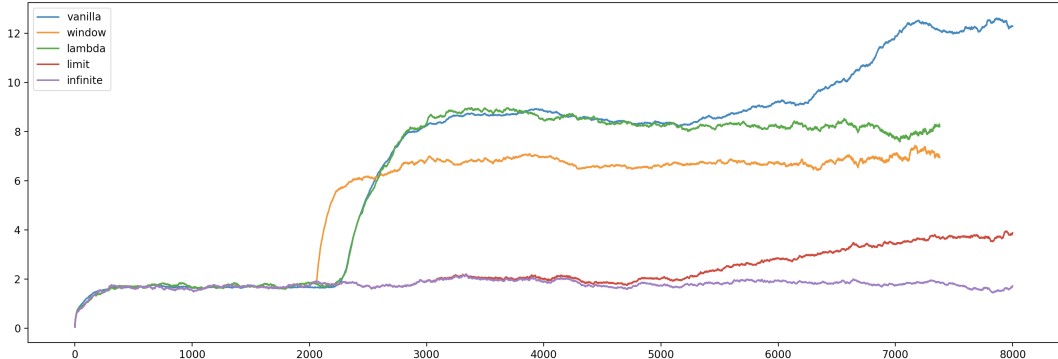

Figure 5: Ablation study by comparing with vanilla Transformer, using windowed attention, using only a Λ-shaped attention mask, and only bounding the attention distance value. Only LM-Infinite achieves stable loss at increasing sequence lengths.

Table 2: Evaluation on text generation on ArXiv and OpenWebText2 corpora. LM-Infinite consistently generalizes the generation quality to extreme lengths, outperforming or similar to the fine-tuned LLM, MPT-7B-Storywriter.

| ArXiv | 4k | | 8k | | 12k | | 16k | | 32k | |
|---|---|---|---|---|---|---|---|---|---|---|
| | bleu | rouge | bleu | rouge | bleu | rouge | bleu | rouge | bleu | rouge |
| MPT-7B-SW | 16.6 | 26.5 | 21.5 | 30.1 | 15.2 | 26.6 | 18.9 | 27.4 | 14.8 | **27.0** |
| MPT-7B | 0.0 | 5.6 | 0.2 | 3.6 | 0.0 | 5.9 | 0.0 | 1.7 | 0.4 | 1.4 |
| MPT-7B + LM-Infinite | 16.1 | 23.8 | 20.2 | 24.9 | 12.6 | 24.1 | 23.9 | 29.0 | **19.7** | 26.6 |
| Llama-2 | 26.6 | 31.4 | 0 | 0.2 | 0.0 | 0.0 | 0.0 | 0.0 | 0.0 | 0.0 |
| Llama-2 + LM-Infinite | **26.9** | **31.8** | **23.6** | **30.9** | **23.9** | **28.2** | **24.8** | **29.2** | 18.4 | 20.4 |
| **OpenWebText2** | bleu | rouge | bleu | rouge | bleu | rouge | bleu | rouge | | |
| MPT-7B-SW | 8.4 | 21.0 | 6.1 | 19.3 | 7.5 | 18.5 | 8.4 | **22.0** | | |
| MPT-7B | 0.9 | 7.5 | 0.9 | 6.6 | 1.0 | 6.4 | 1.0 | 6.8 | | |
| MPT-7B + LM-Infinite | 5.0 | 16.6 | 4.1 | 15.4 | 5.1 | 16.2 | 2.8 | 16.0 | | |
| Llama-2 | 8.8 | **22.4** | 0.0 | 0.2 | 0.0 | 0.0 | 0.0 | 0.0 | | |
| Llama-2 + LM-Infinite | **9.0** | 21.9 | **7.2** | **21.2** | 9.7 | **19.6** | 9.6 | 19.6 | | |

A few more example generations are displayed in Appendix E. We also compare LM-Infinite with a simple baseline of truncating excessive contexts. If one wants to generate texts much longer than the training limit with truncation, frequent truncations and re-encoding of truncated contexts are required. The larger the truncation window $w$ is, the more contextual information is maintained but the larger computation complexity is also incurred. We let the models generate 10k tokens continuously on ArXiv dataset. In Figure 4, it is clear that, with similar computations, LM-Infinite has ∼5 BLEU scores advantage. To achieve similar BLEU scores, LM-Infinite requires only 25% computations than the truncation baseline while still being slightly higher.

## 6 CONCLUSIONS AND FUTURE WORK

In this paper, we provide an explanation and a simple on-the-fly solution to enable Transformer-based LLMs to generate fluently on extreme lengths with relative positional encodings. We start with theoretical and empirical analyses of OOD (out-of-distribution) factors that might contribute to the length of generalization failures. Based on these intuitions we propose LM-Infinite, a plug-and-play mechanism without any parameter updates. Our empirical evaluations show that we can let multiple open-source SoTA LLMs maintain their original generation quality, similar to performance after explicit fine-tuning. LM-Infinite also extends task-solving ability to sequences much longer than training samples. Future work can explore how to let LM-Infinite better perceive information in the masked-out attention region. We hope that LM-Infinite's computational efficiency and ease of use allow researchers without enormous computational resources to also use LLMs on long sequences.

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

## A  PROOF OF THEOREM 1

We first borrow a lemma from Haussler (2018), which we paste below. Note that a cover size $\mathcal{N}(\epsilon, \mathcal{H}, \mu)$ is defined as the minimum cardinal of a cover-set $\mathcal{H}'$ so that any element of $h \in \mathcal{H}$ will be within $\epsilon$ distance to at least one element $h' \in \mathcal{H}'$.

**Lemma 3.** *Let $\mathcal{H}$ be a function family mapping from space $X$ to range $[0, B]$, and its pseudo-dimension $\dim_P(\mathcal{H}) = r$. Then for any probabilistic measure $P$ on $X$, and $\epsilon \in [0, B]$, we have that the $\epsilon$ cover of $\mathcal{H}$ under metric $\mu(h_1, h_2) = \mathbb{E}_{x \sim P}(h_1(x) - h_2(x))^2$ is bounded by:*

$$\mathcal{N}_P(\epsilon, \mathcal{H}, \mu) \leq 2 \left( \frac{2eB}{\epsilon} \ln \frac{2eB}{\epsilon} \right)^r$$

With this lemma we can go on to prove Theorem 1 as follows.

*Proof.* We prove by contradiction. Assume that $\sup_{\mathbf{q}, \mathbf{k}, d \leq n} |w(\mathbf{q}, \mathbf{k}, d)| < a = \left( \frac{\alpha(n)}{2} \right)^{\frac{1}{2r}} \frac{\epsilon}{4e}$. Without loss of generality we can shift all the values to range $[0, 2a]$. Then the function family $\mathcal{H} = \{w(\cdot, \cdot, d) | d \in \mathbb{N}\}$ will have cover size $\mathcal{N}_P(\epsilon, \mathcal{H}, \mu) \leq 2 \left( \frac{4ea}{\epsilon} \ln \frac{4ea}{\epsilon} \right)^r < \alpha(n)$.

However, this is smaller than the number of different distances and relative weight attentions $\mathcal{H}$, which means that at least 2 functions will be close to each other $(w(\cdot, \cdot, d), w(\cdot, \cdot, d'))^2 < \epsilon$. This constitutes a contradiction with the distinguish-ability condition.

$\square$

## B  PROOF OF PROPOSITION 1

*Proof.* Note that entropy on a discrete distribution is defined as $\text{Entropy}(P) = -\sum_i p_i \ln p_i$. Then the attention entropy determined by attention logits $\{w_i | 1 \leq i \leq n\}$ is

$$
\begin{aligned}
\text{Entropy(Attention)} &= -\sum_i \frac{e^{w_i}}{\sum_j e^{w_j}} \ln \frac{e^{w_i}}{\sum_j e^{w_j}} \\
&= -\sum_i \frac{e^{w_i}}{\sum_j e^{w_j}} \left( w_i - \ln \sum_j e^{w_j} \right) \\
&= -\sum_i \frac{e^{w_i}}{\sum_j e^{w_j}} w_i + \ln \sum_j e^{w_j} \\
&\geq -\max_i w_i + \ln(ne^{-B}) \\
&\geq \ln n - 2B \\
&= \Omega(\ln n)
\end{aligned}
$$

$\square$

## C  PSEUDO-DIMENSION ASSUMPTION ON ATTENTION LOGIT FUNCTIONS

In Theorem 1, we assumed that the family of distance-based logit functions $\mathcal{H} = \{w(\cdot, \cdot, d) | d \in \mathbb{N}\}$ has a finite pseud-dimension. In this section, we demonstrate that most current implementations of relative positional encodings do have a finite pseudo-dimension. Let us discuss a few examples in the following:

**T5-Bias and Alibi**  It is easy to see that, the difference between any two logit functions is uniform: $w(\cdot, \cdot, d_1) - w(\cdot, \cdot, d_2) = \text{bias}(d_1) - \text{bias}(d_2)$ regardless of the input. Therefore this family cannot shatter any set larger than 2, so the pseudo-dimension is 1.

**Windowed Attention**  This operation is equivalent to limiting the family to a finite size $|\mathcal{H}| = d_{\max} + 1$, where $d_{\max}$ is the size of the window. Therefore $\dim_P(\mathcal{H}) \leq d_{\max} + 1$.

**NoPE**  As there is no explicit positional encoding implemented, all distances are equivalent. The pseudo-dimension is 1.

**RoPE, CAPE and XPos**  For RoPE, the logit function $w(\mathbf{q}, \mathbf{k}, d)$ is the weighted sum of finite fixed sinusoidal functions $\{\sin(\omega_i d), \cos(\omega_i d)\}$. The size of this set is equivalent to the feature dimension number $k$. We know that $\dim_P(\mathcal{H}_1 + \mathcal{H}_1) \leq \dim_P(\mathcal{H}_1) + \dim_P(\mathcal{H}_2)$. Also, the scaling of a single function can only have pseudo-dimension of 2. Therefore, the whole family has a bounded pseudo-dimension $\dim_P(\mathcal{H}) \leq 2k$. The analysis on CAPE and XPos is similar.

# D  COMPUTATIONAL EFFICIENCY EVALUATION

To evaluate the computational efficiency of LM-Infinite, we run Llama-2-7B model on 100 sequences of 32k length in ArXiv dataset. The hardware is a single A100 GPU with 80GB GPU memory. As the memory is not enough for hosting the whole computation graph during decoding, we use DeepSpeed (Rasley et al., 2020) with Zero3 optimization. We also have to modify the computation code in order to further reduce GPU memory usages and prevent out-of-memory error. With that in mind, the vanilla Llama-2-7B model encodes with an average speed of 48.19s per sequence, while LM-Infinite encodes with average 15.26s per sequences, a 3.16x speedup. The vanilla Llama-2-7B model decodes with 7.34s per token, while LM-Infinite decodes with 2.70s per token, a 2.72x speedup.

# E  EXAMPLE GENERATION

Table 3: Example text generations on ArXiv and OpenWebText2 corpora after 8k context lengths.

| | **ArXiv Dataset** |
|---|---|
| **Context** | *Our proof of the Kottwitz formula imitates these arguments of Langlands and Rapoport of deriving the Kottwitz formula from their conjecture. As such, their theory of Galois gerbs is a major ingredient in this work. Another essential ingredient is a recent work of Kisin [@Kisin17] on the aforementioned conjecture of Langlands and Rapoport. Previously, Kottwitz [@Kottwitz92] proved the formula in PEL-type cases (of simple Lie type $A$ or $C$) by a method which is based on the Honda-Tate theory. This method however cannot be applied in general Hodge-type situations, and indeed* |
| **Vanilla Llama-2** | *of this (of over-equary-her, and [( and, in the...cister '– and an of the model to by. by ..., this, by the. It, and it, 7. –(of an equist (of the. to the [[[WNE (to. and for the (((de in the (for the andistile–c.-[de (for in an inc ort, ort (betness in ¿with (with, based (and (¿ such ((c of a or for the abstract as. of \*. (up. W in the, of the or an. and of a.0 in (and of (over in such [near (—of (([her (pro – [p for [with of a: their in the in of '[alism in the, [(du of that of the ((both [equ of the of [for ([of as of ... and of the of their of ((—for of the of the (t of the of the of the of his of the. of the. of the of it (D with the d. and of of the of the of the of [of of a of a, [[for for a per of a of its of a (for of the and of the* |
| **LM-Infinite** | *Kottwitz [@Kottwitz84a] had formulated the conjecture, but at the time no idea had yet been established on how to prove it. This conjecture involves some algebraic statements of automorphic forms which have an algebraic nature, but are only valid for automorphic forms (which are not holomorphic automorphic forms [@Kaplansky1966]), whose (abstract) objects are Galois extensions of $F$ and $\bar{F}$ (which in general are not necessarily finite extensions), in other words a $p$-valued Hodge-type structure (in $n$-dimension). The Langlands-Rapoport conjecture is now established [@ScholzeShin13] whose formulation is in terms of (a more explicit class of) Hodge-type structures rather than these abstract Galois objects, and the automorphic side for $p$-adic $L$-functions are due to Kisin [@Kisin15].* |

| | **OpenWebText2** |
|---|---|
| **Context** | *This auto-structuration of a complete whole is defined as "auto-regulation". In the individual, the latter is established by biological rhythms, biological & mental regulations and mental operations. These can be theoretically formalized. Piaget refuses to accept that "real" dialectical tensions between physical objects are the true foundations of thought and cognition (its possibility, genesis & progressive development). Piaget never fills in what reality is. He maintains no ontological view on reality-as-such, considered to be the borderline of both the developing subject and its objective world, stage after stage. The cognitive is approached as a process, for rationality grows in developmental stages, each calling for a particular cognitive structure on the side of* |
| **Vanilla Llama-2** | *the in that, of it and and ¿ " of the. of " of " of the pr of of of the of the of the [. of " – s (1 ( of s of the " of quends of it &nd of trail of the, the &, 1. – a and at-and and and the " for on the is, core by " ( as (:: a. -. ( on its to " is a, ( on, th is of all but". ".. in the, in the the the and in the for the. , and of óf the or of the :! of " A. of the of the, of the of the of [of the of the: of a th, and s of the of the the as of the, B of h. the —. the : a, and, in a: & (se and [ ( (t, d of the and....:,. of the (,. ( A. A. Pro: T. I " (b in the óf the, of of: of " of, of the of the [ of, (: - " of the , of s of all ma of the [ and of the of the of [ of the of the of the ( the & ( w [ (:, and ad and and [ " of a in. for the, of ín (* |
| **LM-Infinite** | *cognition. It develops gradually. For Piaget, the cognitive originates in experience as it is subjected to structure : i.e. as it is structured, ordered and transformed. In the first stage, the "sensorial stage", structures are formed and integrated based on perception. These structures are built by organic, biological and psychic transformations, which lead to experience. The "concrete operational stage" of intellectual development refers to a stage when the mind of the child starts to use the logic of the general rule and the rule of the general case. There is now a logical, conceptual and operational distinction of concepts. Reasoning is made explicit by applying logical operations, such as subtraction, addition and multiplication to the "mental" object as well as to the "perceived" world of reality. The child́s logic can now make use of the logical operations, though for him (her) it is a conceptual understanding of abstract operations. This is Piaget́s concept of "genetic development". In the "formal operational stage", logical operations are combined using logical or conceptual structures.* |

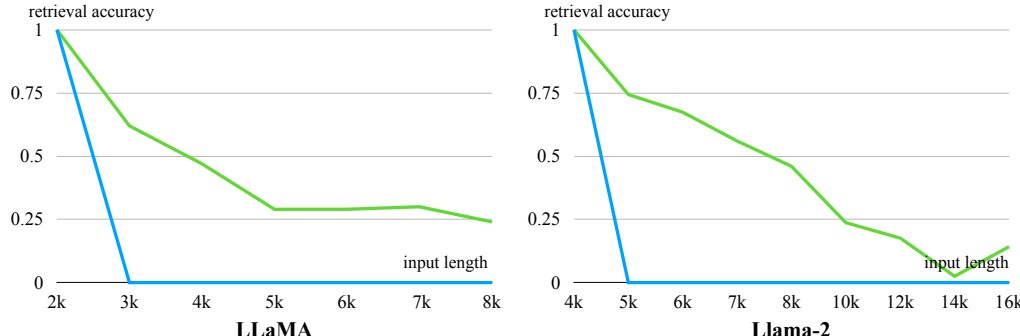

Figure 6: LM-Infinite extends the accuracy on passkey retrieval to longer sequences for LLaMA and Llama-2.

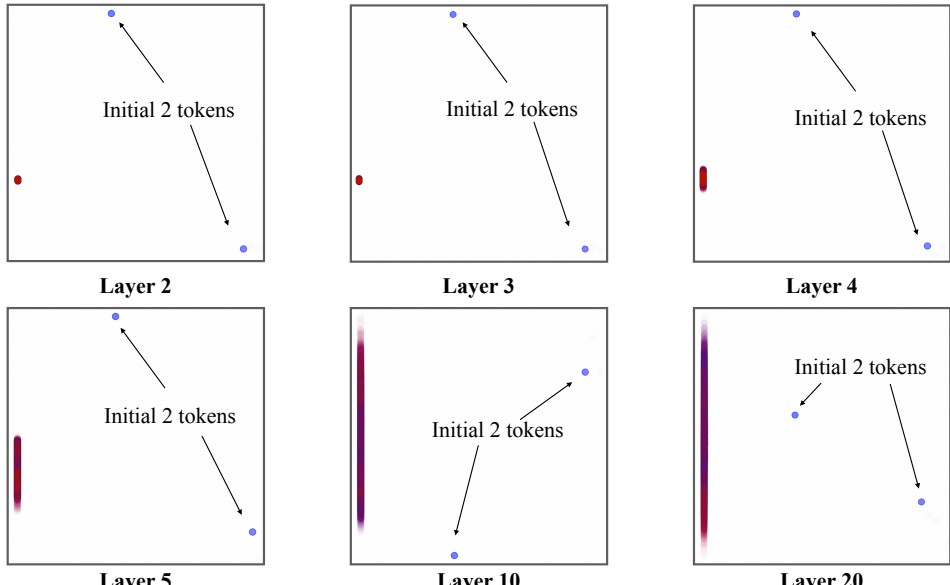

Figure 7: At layers higher than 1, the initial few tokens are implicitly encoded with a strong signal. Abandoning them might cause an OOD factor for the attention mechanism.

## F   TASK SOLVING

Finally, as LLMs are often deployed for downstream tasks, we evaluate how LM-Infinite performs on long-input tasks. We follow  (Mohtashami & Jaggi, 2023) and use the passkey retrieval task. It buries a passkey at a random position in a long distraction text, and in the end, asks what the passkey is. As a synthetic dataset, we have more flexible control on the input length to have fine-grained analysis. We plot the answer accuracy in Figure 6. We can see that LM-Infinite allows LLMs to keep slower decaying accuracy on lengths longer than training, compared to vanilla models which fail immediately. As passkey retrieval is an information-sensitive task, the curve shows that LM-Infinite does not perfectly maintain information perception capability, even though LM-Infinite can theoretically access information from a context as long as $n_{\text{layer}}L_{\text{pretrain}}$. A similar phenomenon, that information in a long document tends to be "lost in the middle" by LLMs, has also been observed by Liu et al. (2023).

## G    MORE ON IMPLICITLY ENCODED POSITIONS

We also plot the token features of higher layers than 1 with PCA projection to the 2D plane in Figure 7. We see that in these layers, the initial few tokens occupy a distinct region with later tokens. Therefore, if these tokens are discarded by window attention during attention, the attention output, which is a weighted sum of $v_i$ vectors, will reside in a different region. This can explain why windowed attention does not work, and why the first few tokens need to be kept by our $\Lambda$-shaped attention.

