# OpenReview forum: "LM-Infinite: Simple On-the-Fly Length Generalization for Large Language Models"
_ICLR.cc/2024/Conference — ICLR 2024 Conference Withdrawn Submission_

### Official Review · Reviewer_M1Yg · 2023-10-28

**Soundness:** 2 fair
**Presentation:** 1 poor
**Contribution:** 2 fair
**Rating:** 3
**Confidence:** 5

**Summary:**

The paper addresses the problem of length generalization failures in large language models (LLMs) when generating long text sequences. The main contributions are:

- Identifying three key out-of-distribution (OOD) factors that contribute to length generalization failures in LLMs: unseen token distances, unseen number of tokens under attention, and implicit encoding of absolute position.

- Proposing LM-Infinite, a simple and efficient solution for on-the-fly length generalization of LLMs. It introduces a Λ-shaped attention mask and distance limit during attention to address the OOD factors.

- Demonstrating that LM-Infinite enables several state-of-the-art LLMs to maintain fluency, perplexity and generation quality on texts up to 128k tokens without any parameter updates or retraining.

- Providing computational analysis showing LM-Infinite has O(n) time and space complexity and leads to 3x faster encoding and 2.7x faster decoding.

Overall, the key contribution is an effective and lightweight technique to improve length generalization in LLMs without costly model retraining.

**Strengths:**

**Originality**: The paper makes an effort to tackle the important problem of length generalization in LLMs. The proposed LM-Infinite technique combines existing ideas like attention masking and distance bounding in a creative way. However, the theoretical analysis of OOD factors draws heavily from prior work.

**Quality**: The paper is reasonably well-written, with adequate experimentation to demonstrate LM-Infinite's capabilities. However, the theoretical analysis lacks rigor in some areas and needs deeper validation. The writing could be improved by explaining concepts and transitions more clearly.

**Clarity**: The overall paper organization follows a logical flow. The introduction motivates the problem and the methods section explains LM-Infinite. But the writing clarity is hurt by a lack of intuition for key equations and insufficient figure captions.

**Significance**: Length generalization is an important challenge for LLMs. By proposing LM-Infinite, this work aims to make some progress on this problem. However, the solution may not fully address the root causes of generalization failures. The technique could benefit from more analysis into why performance degrades at certain lengths.

In summary, while the paper makes some attempts at originality and significance, the technical contributions are incrementally useful rather than groundbreaking. With improved analysis and writing, the work could provide better insights into the length generalization problem.

**Weaknesses:**

1. The writing quality needs improvement for clarity and readability in places:
  - Key equations are stated without sufficient explanation of the terms and implications. Adding intuition would make the technical content more accessible.
  - Figures like 1a, 1b lack explanatory captions to walk through the plots.
  - Overall, several parts of the background and methods lack cohesive flow and transitions between ideas.

2. The motivation for using concepts like pseudo-dimension is not clearly conveyed upfront. The authors should directly state why these theoretical tools are relevant to analyzing the length generalization problem.

3. The analysis of OOD factor 1 reinforces existing intuition that models struggle past fixed training lengths. But it does not provide fundamentally new insights. Existing work such as Chen et al. 2023 has already shown attention logits grow unboundedly at unseen distances.

4. The analysis of OOD Factors 1 and 2 cannot fully explain generalization failures. In Figure 1a, the attention logits do not oscillate much between 5k-15k tokens, yet Llama-2 still fails after 4k. So the explosion in logits alone cannot explain the degradation. Similarly in Figure 1b, entropy increases steadily from 0-30k tokens, but Llama-2 performs well up to 4k tokens. So the growth in entropy alone also cannot explain the failure point. Overall, the increases in attention logits and entropy seem natural to longer sequences, rather than the root cause of failures. The theory may be incomplete by focusing only on these factors. More investigation is needed to fully explain why performance degrades so sharply at a particular length despite no sudden changes in the proposed OOD factors.

5. The claim about initial tokens encoding more absolute position information lacks rigorous validation. The PCA analysis in Fig 1c provides correlational evidence of different subspaces. But further analysis is needed to definitively prove the initial tokens encode absolute position information specifically.

6. More implementation details would aid reproducibility - hardware specifications, dataset preprocessing, method hyperparameter settings etc.

**Questions:**

1. **Reproducibility**: Can you add more implementation details such as dataset preprocessing and method hyperparameter configurations?
2. **Figure 1a Clarification**: What do the different colors of lines in Figure 1a represent, and why does the model collapse at 4k tokens before the attention logits explode?
3.  **OOD Factor 2 - Attention Entropy**: What is the relationship between attention entropy and model performance in OOD factor 2, and why do attention logits steadily increase with the number of tokens in Figure 1b while language modeling performance remains satisfactory up to 4k tokens?
4.  **Figure 1c - PCA Analysis**: What are the implications of the PCA analysis in Figure 1c, and how does it relate to the implicit encoding of absolute position? Can you enhance the analysis to provide a more rigorous validation of the claim that initial tokens encode absolute positional information, moving beyond just showing different subspaces in Figure 1c?
5.  **Pseudo-dimension Analysis**: What is the rationale behind using pseudo-dimension to analyze OOD factor 1, and what are the implications of Theorem 1?
6. **Proposition 1 - Term B**: Can you define or clarify what ‘B’ stands for in Proposition 1?
7.   **Table 1 - LM-Infinite vs. Vanilla Transformer**: Why do the results for LM-Infinite differ from those of the vanilla Transformer within the Pre-training size in Table 1, and can you provide details on the experimental settings of LM-Infinite?

---

> ### Author Response · Authors · 2023-11-23
> **Author Response**
>
> We are glad that the reviewer found the paper “creative” and “reasonably well-written”, and the addressed problem of on-the-fly length generalization “an important challenge for LLMs”. To respond to the comments and questions one by one:
>
> **W1 & 4 & 5 & 7, Q2, 3 & 4** (Clarification of Figure 1 OOD Analysis)
>
> Thanks for the suggestions on clarifying our OOD factor analysis. First, generally, why do LLMs fail faster than logit & entropy explosion? LLMs are usually built on very large models that can have steep local landscapes, subtle perturbations in the features space can lead LLMs to collapse. Instead, our analysis of OOD factor bounds provides a “best case” scenario where LLMs have to fail in the end (a “macro” view). In this way, dealing with these OOD factors constitutes the minimum requirement for LLM length generalization.
>
> Figure 1a Clarification: As stated in Section 3.1 and we will emphasize more in the revision, different colors in Figure 1a are the 0-th attention head in each Transformer layer.
>
> Factor 2 Attention Entropy: entropy introduced by too many tokens provides a potentially hazardous factor that might harm model performance. LLMs are trained on and familiar with the entropy range of fewer than 4k tokens, while the entropy level larger than that constitutes an out-of-distribution zone for them.
>
> Factor 1c: We thank the reviewer for raising suggestions on more direct evidence of OOD factor 3 analysis. We add a plot with PCA analysis of sequence features with higher layers in Appendix G. The results suggest that, at a higher layer than 1, which was plotted in Figure 1c, the initial few tokens are implicitly encoded with a strong signal. Therefore, abandoning them might cause an OOD factor for the attention mechanism.
>
>
>
> **W2, Q5** (Pseudo-dimension Analysis)
>
> Pseudo-dimension is a common assumption for describing the expressiveness of a function family. It holds for most common functions that humans develop, including the relative positional encoding attention functions studied in this paper, with more details elaborated in Appendix C.
>
> **W3** (fundamentally new insights in OOD factor 1)
>
> This factor serves as a motivation to inspire our method proposal. Beyond empirical studies in prior work, here we provide a theoretical explanation of why it has to happen in various position encoding techniques, enhancing the universality of the insight.
>
> **W6, Q1** (dataset preprocessing and method hyperparameter configurations)
>
> Thanks for the suggestion on configuration descriptions. The datasets are tokenized with conventional word tokenizers that are provided along with language models. LM-Infinite is an unparameterized model that relies on very few hyperparameters. The local branch width is selected as the pre-trained length which is the maximum inter-token distance LLMs are applied to. We find that the performance is not sensitive to the global branch width, so the range [10, 100] generally works well.
>
> **Q6** (Proposition 1 - Term B)
>
> $B$ is the value bound for attention logits. Alongside Theorem 1, they provide an OOD dilemma for LLMs: either the logits are bounded and the attention entropy will explode, or the logits themselves will explode.
>
> **Q7** (LM-Infinite vs. Vanilla Transformer)
>
> The difference between vanilla Transformers and those with LM-Infinite is that some long data cannot be passed to vanilla Transformers without CUDA OOM or NaN values. LM-Infinite has an $O(n)$ space complexity and can be evaluated on all data, all with valid values.

---

### Official Review · Reviewer_FyS7 · 2023-10-30

**Soundness:** 2 fair
**Presentation:** 2 fair
**Contribution:** 2 fair
**Rating:** 3
**Confidence:** 5

**Summary:**

The paper highlights that today's LLMs struggle to generate fluent and coherent texts after longer contexts, even with relative positional encoding techniques. Considering that directly fine-tuning on longer corpora is costly, the authors propose LM-Infinite, a simple and efficient solution for on-the-fly length generalization. Technically, they introduce a Λ-shaped attention mask using a distance bound during attention. Experimentally, Infinite-LM yields better generalizability to unseen lengths and provides a decoding speedup compared to existing methods. Empirically, they also conduct a series of analysis experiments to show the reasons of length generalization failure and identify 3 OOD factors.

Contributions:
1. Design experiments and empirically diagnose the factors contributing to generalization failures.
2. Propose on-the-fly decoding method (Λ-shaped attention mask) that provides computational efficiency and generalizability to unseen lengths.

**Strengths:**

1. Identify 3 OOD factors for length generalization.
2. LM-Infinite do not need the fine-tuning or training from scratch.
3. Speed is faster.
4. Consistent fluency and generation quality over longer sequences.

**Weaknesses:**

1. The evaluation is only based on the PPL (Tab1, Fig3) and continue writing (Tab2). However, there is no explicit statement regarding the long-term dependency of LM-Infinite. It raises doubts about whether LM-Infinite can effectively retain useful information in very long chat histories, as the Λ-shaped attention mask design sacrifices some mid-term information. If LM-Infinite falls short in terms of long-term dependency and reasoning, its applicability may be limited.
2. Use of middle tokens. The observation that middle tokens are less position-sensitive aligns with the findings in the paper "Lost in the Middle" [1]. However, completely removing them in LM-Infinite for longer sequences than the training data might not be the optimal solution. It could be more reasonable to address the issue of lost information in the middle, enhancing the model's ability to capture and utilize such information for maintaining long-term dependencies.
3. Unclear OOD factor 3. The analysis of the third out-of-distribution (OOD) factor (Fig1(c)) reveals a distinction between the sub-spaces of initial tokens (blue) and tail tokens (red), offering some evidence that LLMs implicitly encode positional information. However, it is not clear how the conclusion is drawn that the initial tokens contain more crucial positional information.
4. Overclaim about infinite. In Fig3, LM-Infinite demonstrates superior performance compared to the base model. However, beyond 80k tokens, the curves exhibit more noticeable fluctuations, indicating potential limitations of the approach. The claim of "infinite" performance should be approached with caution.

[1] https://arxiv.org/abs/2307.03172

**Questions:**

1. Regarding the positional embedding, as shown in Fig2(b), are the actual tokens $(i-2, i-1, i)$ equal to computing the positions of $(0, 1, 2)$? Or are they computed as *len_of_start_tokens* $+ (0, 1, 2)$?

2. Given $n_{local}=L_{pretrain}$ and $n_{global}\in [10,100]$, the total number of tokens exceeds the pretrained length. Is this a well-designed approach considering the length generalization of LLMs?

3. Are there any experiments conducted to provide evidence supporting the claim made in the conclusion section that "LM-Infinite also extends task-solving ability"?

4. In the statement "3.16x speedup on encoding and 2.72x speedup on decoding," what specifically refers to encoding and decoding?

5. Besides the simple tuncation, can you also compare with the decoding method of Transformer-XL? (as the ablation of the global tokens in the paper)

6. Writing: There are instances of misused citation formats and minor typos present in some places. E.g. (1) Despite extensive explorations in smaller-scale models Press et al. (2021); Sun et al. (2022); Chi et al. (2023) (2) (Press et al., 2021) proposes to offset all attention (3) become "unfamiliar " to LLMs

---

> ### Author Response · Authors · 2023-11-23
> **Author Response**
>
> We thank the reviewer for appreciating the “fluency and generation quality over longer sequences” of the proposed method and the advantage that LM-Infinite does not need fine-tuning. Regarding the comments and questions:
>
> **W1 & 2** (long-term dependency)
>
> We would like to emphasize that this work is the first of its kind for efficiently generating long sequences for existing LLMs without parameter updates. Considering that LLMs tend to get “lost in the Middle,” as the reviewer mentioned, long-term dependency is hard even for LLMs within their pre-training length, let alone an on-the-fly method. So, we leave investigations in this direction for future work, which is out of the scope of the current paper.
>
> **W3** (OOD factor 3)
>
> This factor explains why conventional windowed attention will make LLMs fail, as stated in Section 3.3. The intuitive reason that initial tokens matter is that attention can be imagined as a weighted average (according to attention weights) of the value vectors $v_i$. Windowed attention, however, will reduce the number of seen initial tokens or even discard them as length increases. This will deviate the attention output to “unfamiliar” regions to pre-training scenarios.
>
> **W4** (Claim about infinite)
>
> The reason that curves exhibit more noticeable fluctuations over 80k length is the fewer number of datum that exceeds that length, resulting in a larger variance.
>
>
> **Q1** (Positional embedding)
>
> They are at positions (0, 1, 2), and the initial tokens share position 0.
>
> **Q2** (Approach design)
>
> Thanks for raising this interesting question. We also discuss this intriguing phenomenon in Section 4.1, which reflects the tolerance of the initial tokens by the LLMs. Because they are already very far from the token being generated, their mere existence matters more as long as the number of tokens is not too large (100 << 4096).
>
> **Q3** (task solving)
>
> Thanks for referring to this experiment that we forgot to list in the appendix. We also evaluated the task of passkey retrieval, which requires LLMs to answer a passkey buried in a long context. We follow the setting in Mohtashami et al. (https://arxiv.org/abs/2305.16300). It buries a passkey at a random position in a long distraction text and, in the end, asks what the passkey is. We update the results in Appendix F, which show that LM-Infinite allows LLMs to keep slower decaying accuracy on lengths longer than training, compared to vanilla models, which fail immediately.
>
>
> **Q4** (encoding and decoding)
>
> Typically, LLMs generate from a context or prefix. **Encoding** refers to passing the context or prefix sequence and storing initial key-value caches, which is an expensive process so that the later **decoding** of each token can reuse the cached key-value sequences. As encoding the context and decoding a token has totally different computation costs, we separate the two cases for comparison.
>
> **Q5** (Transformer-XL)
>
> This work focuses on the on-the-fly generation of long sequences for existing LLMs. As Transformer-XL is not compatible with an LLM of different architecture, and forcing it onto another model will make its generation collapse, making a fair comparison is difficult in our setting.
>
> **Q6** (Writing)
>
> Thank you for pointing out the typo. We revised it in the PDF.

---

### Official Review · Reviewer_mkYR · 2023-11-01

**Soundness:** 2 fair
**Presentation:** 3 good
**Contribution:** 3 good
**Rating:** 5
**Confidence:** 4

**Summary:**

This paper studies the length generalization failure on long sequences and identifies three different out-of-distribution factors that contribute to it. Inspired by the analysis of them, the authors propose a simple and effective solution for length generalization that is based on extending the cache of window attention to include the initial tokens without requiring any training. The proposed method is applicable to different length-generalization methods, it is as efficient as window attention and generalizes well to context sizes of up to 128k.

**Strengths:**

- Length generalization in large language models has received a lot of research interest lately. This study proposes a simple method inspired by theoretical & empirical analysis which is interesting. In contrast to most of the existing methods in this area, it does not require any major modification or training.
- The method is applicable to existing length generalization techniques based on relative positional embeddings and it can be potentially impactful as it can be applied to virtually any large language model.
- Experiments on language modeling and text generation show that this method is able to extend the context of different open source models.

**Weaknesses:**

- There is no evaluation of the proposed methods on downstream tasks and models of larger model sizes. The text generation task that it was used is artificially defined. Even though the results are promising, it's unclear how well this method would work in more realistic settings.
- The connection between the theorems and the empirical results were not very precisely made and somewhat hand-wavy; it would be useful to show the theoretical estimate directly in the plot to show how accurate the proven bound is.
- It is not evaluated if the proposed attention method is able to utilize the long context accurately or simply helps with maintaining perplexity at low levels.

**Questions:**

- In the introduction, it would be useful to explain where is the speedup calculated and compare to what method. Also, what is the performance achieved for them?
- The proof of theorem 1 was difficult to follow.  It wasn't clear how a is derived in the proof by contradiction. Can the authors provide more details about the derivation?
- For theorem 2,  shouldn't there be a limit that shows that when n goes to infinity the ln(n) goes to infinity? Also, the entropy of attention will rise to infinity but very slowly given the logarithm.

---

> ### Author Response · Authors · 2023-11-23
> **Author Response**
>
> We are very happy to see that the reviewer finds our work “simple” while “inspired by theoretical & empirical analysis”, and the proposed method “potentially impactful”. In the following, we address your comments and questions one by one.
>
>
> **W1 & 3** (evaluation on downstream tasks, accuracy, and models of larger model sizes)
>
> Our work is focused on extending the generation of language models continuously to longer lengths, which is helpful for tasks like long dialogues and long document generation. In the revised Appendix F, we evaluate the model’s ability to solve tasks by following Mohtashami et al. (https://arxiv.org/abs/2305.16300) and using the passkey retrieval task. It buries a passkey at a random position in a long distraction text, and, in the end, asks what the passkey is. Results show that LM-Infinite allows LLMs to keep slower decaying accuracy on lengths longer than training, compared to vanilla models, which fail immediately. About other model architectures, constrained by academic lab resources, we are only capable of evaluating as large as the size of Llama-2 7B and GPT-J models.
>
>
> **W2** (The connection between the theorems and the empirical results):
>
> Thanks for the suggestions on a clarification of the theoretical part. Figure 1a is hard to verify due to the inaccessibility of constants. However, we are able to add a “bound” curve in Appendix 1b to visualize how the trend is bounded by the theoretical prediction in Proposition 1.
>
>
> **Q1** (Explaining results in Introduction)
>
> The speedup is calculated on the Llama-2-7B model on 100 sequences of 32k length in the ArXiv dataset, with more details elaborated in Appendix D. The comparison baseline is a vanilla Llama-2-7B model. Their performance is the same comparison in Section 4.
>
>
> **Q2** (proof of theorem 1)
>
> Thanks for suggesting clarifications on the proof details. The contradiction is described in the Appendix A. We elaborate it as follows: we first assert a bound on the attention logits within the range [-a, a]. Without loss of generality, we can offset all function values up to [0, 2a] to suit lemma 3 but without affecting other parts of the math. Then for the attention functions on all distances $\mathcal{H}$, lemma 3 tells us that there is a constant bound $\mathcal{N}_P(\epsilon, \mathcal{H}, \mu)$ on the number of “different enough” bins of functions. Here, “different enough” is defined as having an expected distance larger than $\epsilon$. This contrasts with our other assumption that the number of “different enough” bins of functions $\alpha(n)$ grows to let LLMs distinguish newer distance groups. Thus, if $\alpha(n)$ grows, we will need an increasing logit bound $a$ stated in the theorem. We add these explanations in the paper revision.
>
>
> **Q3** ($\ln(n)$ and entropy)
> If we understand correctly, the reviewer refers to our Proposition 1 instead of Theorem 2. Indeed, when n goes to infinity, the ln(n) also goes to infinity, which is an intuition we want to convey in the theorem. Also, the entropy of attention will indeed rise to infinity but very slowly, which accords with our observation in Figure (b). However, we add an ablation study in the paper revision in Figure 5. It shows that this factor leads to a milder OOD hazard compared with other factors, which can be attributed to the slow (but still harmful) growth of attention.

---

### Official Review · Reviewer_sjUS · 2023-11-01

**Soundness:** 3 good
**Presentation:** 3 good
**Contribution:** 2 fair
**Rating:** 5
**Confidence:** 4

**Summary:**

The paper focus on the hot topic of long text/long context of LLM. It theoretically and empirically investigates the main out-of-distribution (OOD) factors contributing to this problem and proposes a solution for length generalization for LLMs with relative positional encodings.

**Strengths:**

This paper has the following advantages in solving long text problems:
1. It diagnoses the OOD factors when the length of sequences is longer than the training time. It provides adequate theoretical analysis and experimental verification, which probes directions to optimize the length generalization.
2. A plug-and-play decoding method, LM-Infinite is proposed to be applicable to unseen lengths without parameter updating and finetuning to some extent.

**Weaknesses:**

1. Lack of originality and significance. While the theoretial analysis in this paper is a splotlight as an instruction for further optimization, the techniques of LM-Infinite by combining both global and local attention under limit distance has been proposed before in many similar works such as Longformer, LongNet, etc. Those methods employed in recent long context LLMs have shown outstanding performance in solving long text and need further discussion and comparisons in this paper.

2. Few experiments and ablation studies. More analysis is required on the experiments results to probe more insights on this topic. Besides, there are less comparisons with existing state of arts models and techniques targeting to solve similar problems. Furthermore, it is recommended to assess the models' performance in real multi-task cases with delicately selected corresponding metrics for evaluation instead of automatical n-gram metrics with bias.

3. Dataset used in the experiments are far from adequate for evaluating long context processing and length generalization. Arxiv paper and OpenWebText2 are too domain specific with specific characteristics and cannot applied to evaluate the real capability of LLM in the downstream tasks encountering long text. Please refer to the well acknowledged datasets that are recently proposed for long context evaluations such as ZeroScrolls, LEval, Longbench and etc.

**Questions:**

See more in Weaknesses sections.

---

> ### Comment · Reviewer_sjUS · 2023-11-23
>
> So far I didn't see a clear tendency to provide a rebuttal or update the work. Considering the current version, I tend to reject this work.

---

> ### Author Response · Authors · 2023-11-23
> **Author Response**
>
> First of all, thank you very much for appreciating the “adequate theoretical analysis and experimental verification” in the paper and the “plug-and-play” nature of the proposed method. Regarding your comments and questions:
>
> **W1** (Lack of originality and significance):
>
> In this paper, we focus on understanding and applying these techniques in a totally different direction: on-the-fly adaptation of LLMs to longer generations. In contrast to our method, most previous papers focus on training from scratch or fine-tuning longer texts for long generations. This demand for such fine-tuning will be endless if the length of inputs continues to increase. For this reason, our papers are under different settings. More importantly, our paper provides more insights into what position representation was learned during pre-training, which is of larger value for the NLP community.
>
> **W2** (Few experiments and ablation studies):
>
> Thank you for this constructive suggestion. In the paper, we evaluate generation quality and decoding speed compared with baselines. We agree that ablation studies can give more insights into why alternative techniques might fail, so we compare with the baselines of vanilla Transformer, using windowed attention, using only a $\Lambda$-shaped attention mask, and only bounding the attention distance value. The result is added as Figure 5 in the revision. We see that all these baselines fail to maintain stable loss when length increases. This validates that it is essential to combine the two proposed components for LLM length generalization.
>
>
> **W3** (Dataset used):
>
> Thank you for your interest in future exploration in this direction. We would like to point out that the current work focuses on the problem of “length representation generalization” of large language models (LLMs). On the text generation task, the proposed method demonstrates dramatic improvement by allowing LLMs to generate at 32k (much longer than pre-training length) with stable quality. To evaluate the model’s ability to solve tasks, we follow Mohtashami et al. (https://arxiv.org/abs/2305.16300) and use the passkey retrieval task. It buries a passkey at a random position in a long distraction text and, in the end, asks what the passkey is. We update the results in Appendix F, which show that LM-Infinite allows LLMs to keep slower decaying accuracy on lengths longer than training, compared to vanilla models, which fail immediately.